# Impact on Sleep Quality, Mood, Anxiety, and Personal Satisfaction of Doctors Assigned to COVID-19 Units

**DOI:** 10.3390/ijerph19052712

**Published:** 2022-02-25

**Authors:** Pilar Andrés-Olivera, Judit García-Aparicio, María Teresa Lozano López, José Antonio Benito Sánchez, Carmen Martín, Ana Maciá-Casas, Armando González-Sánchez, Miguel Marcos, Carlos Roncero

**Affiliations:** 1Psychiatry Service, University of Salamanca Healthcare Complex (CAUSA), 37007 Salamanca, Spainjabenito@saludcastillayleon.es (J.A.B.S.); mcmarting@saludcastillayleon.es (C.M.); amacia@saludcastillayleon.es (A.M.-C.); croncero@saludcastillayleon.es (C.R.); 2Psychiatric Unit, School of Medicine, University of Salamanca, 37007 Salamanca, Spain; 3Institute of Biomedicine of Salamanca (IBSAL), 37007 Salamanca, Spain; jgarciaa@saludcastillayleon.es (J.G.-A.); mmarcosm@saludcastillayleon.es (M.M.); 4Internal Medicine Service, University of Salamanca Health Care Complex (CAUSA), 37007 Salamanca, Spain; 5Medical Department, School of Medicine, University of Salamanca, 37007 Salamanca, Spain; 6Department of Statistics, University of Salamanca, 37007 Salamanca, Spain; armando_gonzalez@usal.es

**Keywords:** COVID-19 pandemic, doctors, mental health impact

## Abstract

The SARS-CoV-2 health emergency has led to a restructuring of health care systems and the reassignment of medical specialists from their usual duties to attend COVID-19 patients. The aim of this paper is to describe the levels of insomnia, anxiety, depression, and the impact on quality of life of doctors who were on the frontline of COVID-19 during the first two waves of the pandemic. Self-report surveys were conducted on said physicians during both waves, with 83 and 61 responses in the first and second waves, respectively. The reported presence of insomnia was frequent (71.8%), although it decreased in the second survey. Anxiety was moderate, decreasing from 57.1% to 43.1% between measurements. Overall, depression rates decreased between the two surveys. Substance use was found to have an indirect correlation with personal and professional satisfaction. In the light of the unforeseeable evolution of the pandemic and the medium- to long-term repercussions on professionals, we believe the adaptation of health resources is crucial to meet the new unpredictable mental health needs of this group.

## 1. Introduction

In March 2020, a state of alarm was declared in Spain due to the COVID-19 pandemic, unpredictably transforming the social and economic environment, as well as health care systems [1]. All these elements contributed to environmental stress in everyday situations. In addition, health professionals found themselves in an unprecedented situation that forced organizational restructuring of the social and health care services in the context of limited information and high levels of stress [2,3,4].

In the context of this pandemic, health professionals have been shown to suffer from psychological consequences such as insomnia, anxiety, depression, or stress [1,4,5,6,7,8,9,10,11], especially when they perceive a lack of resources and support [4], as well as higher levels of optimism and hope [12]. In the pandemic situation, compassion fatigue and burnout levels have remained moderated or high among healthcare professionals [13].

The presence of insomnia has been linked to several pathologies such as hypertension [5] or dissatisfaction with life. Similarly, a correlation between insomnia and depression has been found [6].

In Spain, typical levels of poor sleep quality are around 38.2% [14], with sleep disorders affecting approximately 30% of the general population [15]. According to some studies, 30% of health care professionals caring from COVID-19 patients suffered from poor sleep quality [7], 71.6% of them from anxiety, and 60.3% from depression [8].

The main objective of this study is to describe the levels of insomnia, anxiety, depression, and the impact on quality of life of doctors who were on the frontline of COVID-19 care.

A secondary objective of the study is to evaluate use of substances as way of coping with difficulties of adapting to a different way of working in the pandemic context. Another objective is to classify the participating physicians into groups of greater or lesser intensity of the symptoms studied.

The quality of sleep among the sample of doctors in this study will also be compared with another study conducted in other Spanish Hospital to see if the impact of the pandemic has been similarly affected.

## 2. Materials and Methods

### 2.1. Participants and Research Desing

The sample consisted of doctors who became part of the frontline care teams (referred to as COVID-19 units) and were reassigned to coronavirus hospital wards to provide direct attention to patients, excluding emergency and ICU physicians, as no COVID teams were created in these areas; 70% were women, and by specialties, 29% were internal medicine specialists and the rest were from all the other medical specialties working at the hospital. Ages ranged from 25 to 62 years old. These teams were composed of professionals from diverse medical specialties, including those treating similar pathologies to SARS-CoV-2 (internal medicine and pneumology), but also others with little-to-no experience with this type of pathologies (surgical, medical-surgical, central services, and other medical specialties). The survey was sent to 110 physicians reassigned to the COVID-19 units during the first wave, and 83 took the survey (75.45% of the staff). In the second wave, the survey was sent to the same group of professionals, obtaining 61 responses (55.45%). However, only some of them were assigned back into the COVID-19 units, while the rest returned to their usual specialties.

Two cross-sectional studies were carried out at two different times, coinciding with the first and second waves and with two similar samples.

The University of Salamanca Health Care Complex was composed of three different hospitals at the beginning of the pandemic. During the first wave, one of them treated only patients with pathologies unrelated to COVID. The other two were set up for the care of SARS-CoV-2 diseases. The clinical hospital provided 12 of its 14 wards available. In the second wave, only half of these wards were needed. The care of these patients admitted for COVID was carried out only by the physicians to whom the survey was sent. The rest continued with their usual work with the necessary adaptations.

### 2.2. Survey Instrument

The invitation to participate in the study was sent out electronically on both occasions, using an official mailing list that had been set up during the first wave of the pandemic for the distribution of information to COVID-19 units. The Ethics Committee was informed in advance, and registered under the Ethics Committee code number 2020 06 500. Responses were collected from 20 to 26 April 2020, partly coinciding with a decrease of COVID-19 cases during the first wave (Figure 1). The second survey took place between 7 August 2020 and 11 November 2020, coinciding with the rise, peak, and fall of the second wave of COVID-19 cases.

The questionnaire sent to the participants was available on the Google Forms platform. It included the Spanish adaptations of the Insomnia Severity Index (ISI) [16], the Beck Anxiety Inventory (BAI) [17], the Beck Depression Inventory (BDI-II) [18], and an ad hoc questionnaire on life satisfaction, as well as a questionnaire on substance use that was only included in the second wave.

The Spanish version of the ISI [16] is an improved translation [19] of the original questionnaire [20]. This scale consists of seven items: the first three are related to sleep difficulties; item 4 corresponds to sleep satisfaction; and items 5, 6, and 7 to the impact of insomnia. The thresholds of the scale are as follows: no clinically significant insomnia (0–7), subthreshold insomnia (8–14), clinical insomnia (moderate severity) (15–21), and clinical insomnia (severe) (22–28).

The BAI [17] has been widely used as a screening test in the general population. This scale provides an overrepresentation of the somatic dimensions of anxiety, and is consistent with the DSM-IV diagnostic criteria [21]. The correction was based on the sum of the values of those items, ranging from 0 to 3, according to their degree of severity. In a non-clinical Spanish population, a score of 12 is accurate for diagnosing 81% of cases of anxiety disorder, reaching 90% of accuracy when the score increases to 19 [22]. However, the inventory for the Spanish population suggests using a range of 0 to 9 for non-anxiogenic, 10 to 18 for moderate anxiety, 19 to 29 for moderate to severe anxiety, and over 30 for severe anxiety [17]. It presents a bifactorial structure: the first factor evaluates somatic symptoms, and the second one evaluates subjective anxiety and panic symptoms.

Depression was assessed according to the second version of the Beck Depression Inventory (BDI-II) [22,23,24,25], i.e., the Spanish adaptation of the original 1961 test [26]. It includes two factors: the cognitive-affective and the somatic-motivational factors. Normative data on general population were obtained from the BDI-II validation study [23]. The thresholds of the scale are minimal depression (0–13), mild depression (14–19), moderate depression (20–28), and severe depression (29–63)

The ad hoc questionnaire consisted of items specifically designed for this study, comparing the current situation with the tasks performed before the reassignment in COVID-19 units. In both surveys, satisfaction was rated on a scale from 1 (dissatisfied) to 5 (very satisfied). The substance use questionnaire assessed the use of benzodiazepines, tobacco, and other substances, as well as regular alcohol consumption. This scale used the values of “never used”, “started”, “decreased”, “stable”, or “increased”.

The insomnia results of the study sample were compared with another sample of health professionals (no only doctors) from the Health Care Complex of the Hospital 12 de Octubre and a control group [27].

A *p*-value < 0.05 was considered significant. Means and standard deviations (SD) were used to analyze the descriptive statistics, as well as percentage differences and coefficients of variation (CV). Cohen’s d, Hedge’s g, Δ2 Glass, and R^2^ were also used as indicators of effect size. For the analysis of insomnia, our sample was compared with a very similar sample made up of health professionals (doctors and others) from the 12 de Octubre hospital in Madrid and people whose jobs were not health care (control group). For the analysis of the correlations between variables, bivariate correlations using Pearson’s correlation coefficient were used. The multivariate analysis of satisfaction, anxiety, depression, and insomnia was performed using principal component analysis of HJ-Biplot [28] and the MultBiplot software, 09-01-2020 version [29]. Additionally, SPSS v.25 [30] was used for the remaining analyses.

In order to classify respondents, segments were formed according to the different mental health conditions. For the segmentation analysis, only those respondents who answered all the questions were included. The examined variables were insomnia, anxiety, and depression, with a new variable created to integrate them. From the latter variable, three segments were created using Ward’s method. Substance use was contrasted with the severity of the mental health condition, in order to analyze its impact on people’s behavior.

For the segmentation analysis, only those respondents who answered all the questions were included. The examined variables were insomnia, anxiety, and depression, with a new variable created to integrate them. From the latter variable, three segments were created using Ward’s method. Substance use was contrasted with the severity of the mental health condition, in order to be able to analyze its impact on people’s behavior.

## 3. Results

The sample was composed solely of physicians whose professional practice was carried out in any unit of Salamanca Hospital. The overall results are presented in the following table (Table 1)

### 3.1. Sleep

The results obtained in the first wave revealed the presence of mild insomnia, while the second wave showed a clinical absence of insomnia.

The prevalence of insomnia during the first wave was high (71.8%), decreasing to 50.9% in the second wave. No doctors were affected by severe clinical insomnia. ISI scores went from a mean value of 10.91 (SD ± 5.46) to 7.88 (SD ± 2.89) (d = 0.66). Overall, there was a decrease in the severity of sleep difficulties (including sleep onset, maintenance, and early morning) and the distress caused by sleep problems. There was an increase in the interference of sleep problems with daily functioning and sleep dissatisfaction.

As for the perceived external noticeability of this problem in daily life, the results showed a decrease in extreme values, while the intermediate values increased. The quality of sleep improved (from mean 10.91 to 7.88), reaching similar values to those of other samples of health workers (not exclusively doctors) (mean = 7.83) [27], but not as high as those of the general population (mean = 6.32). The results indicated that the quality of sleep of doctors increased between the first and second wave (Table 2). There was a decrease in the moderate severity of sleep difficulties, while the percentage of no clinically significant insomnia and subthreshold insomnia increased (Table 2).

The severity of insomnia decreased in the dimensions of sleep difficulty (onset, maintenance, and awakening) and satisfaction. However, the impact of insomnia increased in terms of interference with daily functioning. Sleep-related distress significantly diminished, while external noticeability of sleep problems clustered at intermediate scores (Table 3).

The results were compared with the sample from the Hospital 12 de Octubre in Madrid (Spain), which was constituted by COVID-19 frontline healthcare workers, including nurses, auxiliary nurses, orderlies, and cleaning staff. In the first measurement of our study, insomnia was significantly higher than in the control group, and moderately higher than the general healthcare sample of the Hospital 12 de Octubre. In the first measurement of our sample, 30.8% of participants presented moderate severity insomnia. When compared to the control group, the effect size obtained was considerable [31].

The values obtained in the second survey were close to those of the general healthcare sample of the Hospital 12 de Octubre. The differences between this second measurement and the control group showed a medium effect size (Table 4).

### 3.2. Anxiety

Anxiety was moderate in both the first wave with a mean value of 14.32 (SD ± 10.30), and in the second wave with a mean value of 11.58 (SD ± 7.38). Scores were obtained for all symptoms, with the most severe item being “difficulty in breathing”, while the item “feeling hot” was the only one that increased in severity (Table 5).

Comparing the first and the second wave, anxiety decreased from 57.2% of participants classified as anxious to 43.10%, with a small to medium effect size (d = 0.29). Subjective anxiety symptoms and panic symptoms predominated over somatic anxiety on both measurements.

Anxiety factors presented differences with a small to moderate effect size (d = 0.33), showing a higher prevalence in subjective anxiety and panic symptoms than in somatic anxiety (Table 6).

### 3.3. Depression

Depression scores were low, with a mean value of 14.76 (SD ± 7.51) in the first wave, and minimal levels with a mean of 11.80 (SD ± 8.74) in the second wave. In both waves, the highest scoring items were the changes in sleep patterns, fatigue, loss of energy and difficulty concentrating, whereas the least frequently displayed and mild symptoms were the thoughts about suicide and punishment. However, feelings of sadness and failure were found to be higher in this sample than in other general population samples [24,32,33] (Table 7 and Table 8).

Compared to the norm, the first measurement showed a large effect size (d = 0.88), while the second measurement showed a medium effect size (d = 0.46) [23]. When comparing the two measurements, there was a reduction in depression with a medium size (d = 0.36). While most depressive symptoms decreased, the following items increased: pessimism, indecisiveness, irritability, self-dislike, suicide (five more people in this item), and punishment.

### 3.4. Satisfaction

Personal and job satisfaction of doctors mostly consisted of “average” and “slightly above average” values, with mean scores of 3.47 (SD ± 0.82) in the first wave and 3.34 (SD ± 0.76) in the second wave (Table 9).

When comparing both measurements, the average scores for the current personal and job satisfaction increased, while the extreme scores decreased. Overall, the participants’ satisfaction increased.

The number of people who reported to be “not very satisfied” or “dissatisfied” decreased, i.e., their level of satisfaction increased. The amount of people who stated they were “very satisfied” also decreased. Consequently, the number of people who reported intermediate scores of “normal” or only “satisfied” on a personal level increased. These changes in extreme values affected the means of the first and second wave, making the difference between them minimal in terms of personal satisfaction (d = 0.02), and small for previous job satisfaction (d = 0.29). Nevertheless, there were changes within the sample, even if they were not reflected in the mean.

### 3.5. Substance Use

The levels of substance use were low. Most respondents reported not using benzodiazepines (68.9%), alcohol (75.4%), or tobacco (86.9%); no other substance was mentioned in the survey. Lower personal satisfaction was found to be associated with a higher use of benzodiazepines (r = −0.30; *p* = 0.01). Furthermore, a correlation was found between alcohol consumption and job satisfaction: the lower the job satisfaction (both before (r = −0.38, *p* = 0.002) and after (r = −0.37, *p* = 0.006) reassignment to COVID-19 units) the higher the alcohol consumption (Table 10).

### 3.6. Correlation between Variables 

The results show that higher levels of job satisfaction were linked to a better quality of sleep and lower levels of anxiety; no correlation was found between the previous working situation and sleep or anxiety (Table 11). Similarly, no differences were found between regular alcohol consumption and the working situation before or after participants were reassigned to COVID-19 units. Regardless of whether it referred to the current or previous position, a higher job satisfaction implied lower alcohol consumption. On the other hand, depression was inversely related to personal satisfaction, as well as the employment situation (both the previous and current position); depression is positively related to anxiety (r = 0.69; *p* < 0.001), insomnia, and the use of benzodiazepines. No correlation was found between depression and tobacco or alcohol use.

#### 3.6.1. Multivariate Analysis

As shown in Figure 2, axes 1–2 show 70.2% of the variance, making the HJ-Biplot a good indicator to assess the variability of scores on depression, anxiety, job, and personal satisfaction and insomnia.

The acute angle formed by the two factors of the BAI anxiety scale show that they were closely related to each other. Anxiety factor 1 (somatic symptoms) was more strongly related to sleep, while factor 2 (subjective anxiety and panic symptoms) had a closer correlation with depression.

The variables of depression, anxiety, and sleep were found to be independent in relation to job satisfaction before respondents were assigned to COVID-19 units. Our analysis revealed an inverse correlation between the variables of depression, anxiety, and sleep with personal and job satisfaction during assignment to COVID-19 units. The said correlation was stronger in the case of the depression variable and weaker in the sleep variable (Figure 2).

#### 3.6.2. Segmentation of the Sample According to Psychological Dimensions and Their Characterization

Three groups were created for the segmentation analysis, classifying them according to mental health severity as follows: cluster 1, minimal mental health distress; cluster 2, medium mental health distress; and cluster 3, extreme mental health distress (Figure 3).

Differences were found between the two measurements, with an increase in clusters 1 and 2 and a decrease in cluster 3. The number of people with extreme mental health distress decreased, while the number of people who were reported to have minimal to medium mental health distress increased (Table 12).

Cluster 1 was constituted by people who reduced their use of benzodiazepines or did not use them at all, while clusters 2 and 3 represented those respondents using benzodiazepines. As can be seen, the percentage of clusters 2 and 3 decreased between measurements. This was consistent with the correlation of benzodiazepine use with sleep, depression, and anxiety.

Alcohol and tobacco use were not related to anxiety, depression, or insomnia; therefore, the correlation with the clusters was non-existent.

As satisfaction was negatively correlated with insomnia, anxiety, and depression, the respondents belonging to cluster 1 had higher sample percentages in the options “satisfied” or “very satisfied”. Meanwhile, the level of satisfaction decreased in cluster 2, and was even lower in cluster 3.

## 4. Discussion

There was a reduction in the severity of all variables (insomnia, anxiety, and depression) between the two measurements in our study. This study allowed us to measure the evolution of the variables (insomnia, anxiety, and depression) between the two measurements in a group of doctors who were assigned to the COVID-19 units during the first wave.

The presence of insomnia was significant (71.8%), with a decrease in insomnia levels in the second wave, which had a tendency towards homogeneity. Insomnia was correlated with depression and anxiety. No major psychopathological alterations were detected as a result of this forced reorganization, although there were significant effects on the quality of sleep.

Anxiety was moderate, decreasing from 57.1% to 43.1% between measurements. Overall, levels of depression decreased, but they did increase in certain cases that presented severe symptoms, such as the small group with suicidal thoughts. This said group was detected in only one item, although it was considered well-founded on its own as a measure for suicidal ideation [32] and was a valid pathological indicator [33]. Depression had a correlation with personal and job dissatisfaction. There was no observed correlation between COVID-19 care and depression or anxiety. The source of stress was rather attributed to the uncertainty of not knowing when the pandemic would be under control [34].

A decrease in the variable scores was observed, which could be due both to the fact that a certain number of doctors did not participate in the COVID-19 units in the second wave, and to a more effective management of the situation. This said improvement could be the result of the following elements: the different organization of care shifts, less hardship, a broader knowledge of the situation, reduced distress, better knowledge of the disease, and a reduced risk of contagion.

The use of substances was found to have an indirect correlation with personal and professional satisfaction.

Even though personal dissatisfaction was correlated with insomnia (R^2^ = −0.26 **), it should be noted that the dissatisfaction-insomnia correlation was unidirectional. That is, insomnia does not produce dissatisfaction with life, but dissatisfaction with life does produce insomnia [35].

A correlation was found between benzodiazepine use and depression, insomnia, and anxiety (even more closely related to factor 2 of subjective anxiety and panic symptoms). There was also a correlation between alcohol use and job dissatisfaction, but no correlation was found between tobacco use and job dissatisfaction in any of the variables studied. Increased use of benzodiazepines was also reported in the general US population during the pandemic. In the case of doctors, this use seemed to be linked to the appearance of psychopathological distress.

A high prevalence of insomnia and moderate levels of anxiety and depression were identified, as they had already been described in other studies conducted at the beginning of the pandemic [11].

Differences were detected with other similar samples (higher levels of insomnia), although this could be explained by the type of professionals that participated in our study, who were exclusively doctors [36].

The values of anxiety were consistent with the ones obtained in the samples of COVID-19 frontline healthcare professionals (43%) [36].

Although substance use can be considered a maladaptive type of resilience that is more prevalent in males [37], no significant increase in substance use was found compared to pre-pandemic levels.

This is a real-life study that could potentially help to understand the real professional performance of doctors. However, the second survey was conducted during the second wave. At this point in time, most doctors were no longer in COVID-19 units but were performing their usual tasks. However, unlike in the general population, symptomatology was not reduced in this group. This could be a result of the uncertainty surrounding the issues of disease control and its effectiveness.

One of the limitations of this study is a sampling bias, as it is possible that the subjects who responded to the survey on the two occasions were different, which could mean that they were self-selected, with, for example, the most affected subjects not responding.

Another limitation of the study is that the two times the survey was sent out, although the responses in the first wave were available for the same amount of time, they were recorded in the first week, while in the second wave, they took longer to be recorded, which may have implications for comparisons.

We were not able to ask each participant their sex, age, or specialty, as these data would have allowed us to identify the respondents, which is a limitation for interpreting results.

While this study has the potential to help understand the real challenges of medical practice, it does have limitations in terms of generalizability of the results, like with other naturalistic studies. There were several elements that differentiated the first wave from the second: initially, there was greater scientific-technical lack of knowledge regarding medical decisions, uncertainty about the effectiveness of political decisions on the epidemic, absence of historical references in our immediate environment, prolonged home confinement, etc. These factors, together with the employment situation of the respondents, hindered the identification of specific causal attributions. Furthermore, the study was exclusively limited to those doctors in the COVID-19 units working in the University of Salamanca Health Care Complex during the pandemic, so the results shown may not be fully applicable to other contexts or population groups.

However, the fact that the general population still maintained high COVID-19 symptomatology may be due, among other factors, to the reorganization of COVID-19 units. Thus, it is possible that our data are not directly applicable to other hospitals, which may have structured COVID-19 units differently, or did not experience such an acute first wave, as it was the case in our province.

This study could be extended to second-line professionals who have not treated COVID-19 patients directly, but whose routine has also been affected by this situation.

It would be interesting to study the psychopathological consequences of traumatic manifestations during the following waves of the pandemic, as well as its long-term evolution.

## 5. Conclusions

This study shows that it is necessary to implement preventive and therapeutic measures in order to detect mental health problems. This forced medical reorganization did not cause any major psychopathological distress, only affecting the quality of sleep. However, we must bear in mind that the statistics can “hide” significant health data (almost 8% of the doctors in the first wave sample presented severe anxiety scores (Table 5) and almost 7% had severe depression (Table 8). In addition, measures with a central tendency could possibly counteract scores on uncollected variables (e.g., no differentiation was made between professionals who volunteered and those who felt forced to take on the reassignment).

However, the severity decreased in the second measurement, coinciding with the second wave of the pandemic, when most of the COVID-19-unit members remained in their usual position, although their working conditions had been partly modified by the pandemic. Another study described a decrease in the intensity of symptoms, although it pointed out the lack of consistency due to the uncertainty surrounding the situation, warning of the possible development of irritability, hypervigilance, or nightmares derived from the experience [38]. Fear, uncertainty, and stigmatization are recurring elements in every biological disaster and can become an obstacle to provide adequate medical and mental health treatment [39].

Although we observed high scores on the scales, they were not as high as in other studies conducted later; possibly one of the factors that influenced this was that the reorganization of attention was effective.

As a final conclusion, this study shows that the psychological impacts studied generally decreased between the two surveys. However, in a population with presumably available coping resources, consequences may manifest in other dimensions such as grief, trauma, or guilt, as well as traumatic reactions that may resurface when recalling participation in COVID-19 units. Nevertheless, feelings of sadness and failure were higher in this sample than in other general population samples [24,32,33].

Given the uncertainty of how the pandemic will actually evolve and when it will end, as well as the medium- and long-term repercussions on professionals, we believe it is important to adapt health resources to the new and unpredictable mental health needs of this group.

## Figures and Tables

**Figure 1 ijerph-19-02712-f001:**
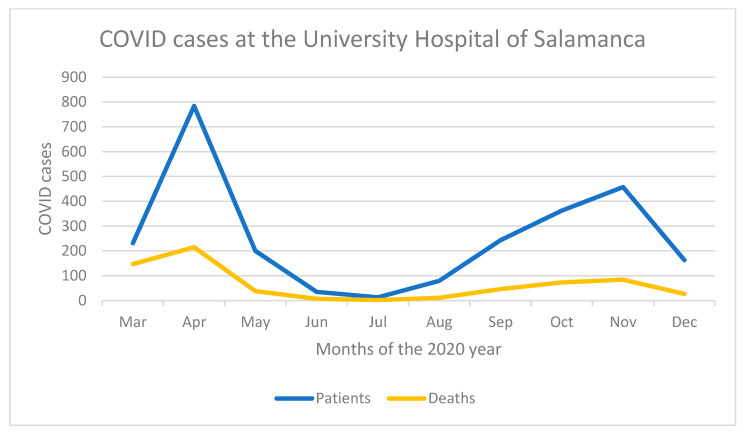
Evolution of COVID-19 cases at the Clinical Hospital of Salamanca for 2020.

**Figure 2 ijerph-19-02712-f002:**
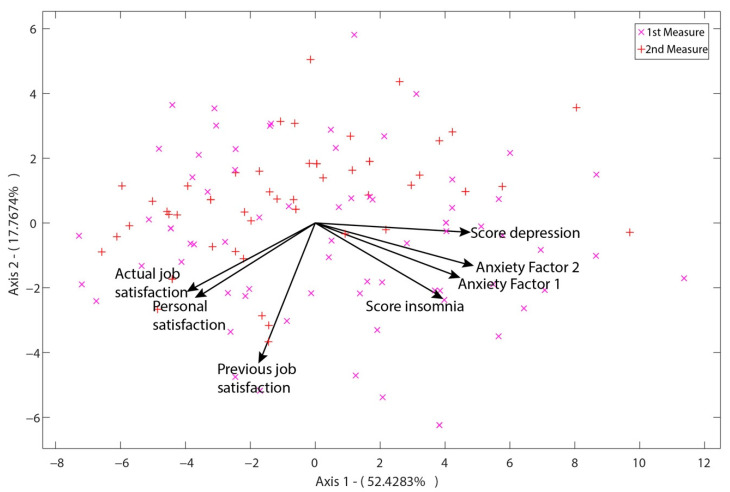
Two-dimensional representation of the HJ Biplot analysis using axes 1–2. Barycentric scaling. First measurement (x), second measurement (+).

**Figure 3 ijerph-19-02712-f003:**
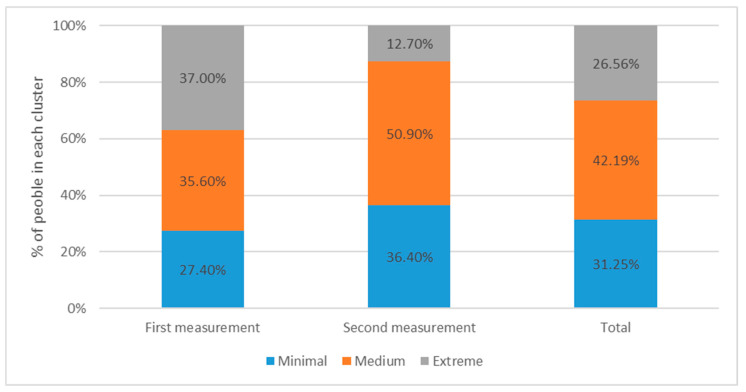
Measurement-based segmentation.

**Table 1 ijerph-19-02712-t001:** Summary of scale scores between waves.

	1st Wave	2nd Wave
ISI (+8)	71.8	50.8
BAI (+10)	57.1	43.1
BDI (+14)	46.7	58.3

Note: percentage of sample who scored that punctuation or more, which means significant results.

**Table 2 ijerph-19-02712-t002:** Frequency of ISI scores.

ISI Results	1st Meas.		2nd Meas.		Dif %
	*n*	%	*n*	%	
No clinically significant insomnia (0–7)	22	28.2	29	49.2	21.00
Subthreshold insomnia (8–14)	32	41.0	29	49.2	8.20
Clinical insomnia (moderate severity) (15–21)	24	30.8	1	1.7	−29.10
Clinical insomnia (severe) (22–28)	0	0	0	0	0
Total	78	100	59	100	

**Table 3 ijerph-19-02712-t003:** Frequency of ISI dimensions.

	Sleep Difficulty																			
	Onset					Maintenance					Early Morning									
	1st Meas.		2nd Meas.		Dif %	1st Meas.		2nd Meas.		Dif %	1st Meas.		2nd Meas.		Dif %					
Intensity	*n*	%	*n*	%		*n*	%	*n*	%		*n*	%	*n*	%						
None	14	17	17	28	11	17	21	19	31	11	21	25	23	38	12					
Mild	18	22	27	44	23	23	28	18	30	1.8	21	25	20	33	7.5					
Moderate	42	51	17	28	−23	34	41	24	39	−2	22	27	13	21	−5					
Severe	8	9.6	0	0	−10	9	11	0	0	−11	15	18	3	4.9	−13					
Very severe	1	1.2	0	0	−1	0	0	0	0	0	4	4.8	0	0	−5					
n	83	100	61	100		83	100	61	100		83	100	59	100						
	**Impact of Insomnia**															**Dissatisfaction with Sleep Quality**				
	**Interference**					**Noticeability**					**Distress**					**Satisfaction**				
	**1st Meas.**		**2nd Meas.**		**Dif %**	**1st Meas.**		**2nd Meas.**		**Dif %**	**1st Meas.**		**2nd Meas.**		**Dif %**	**1st Meas.**		**2nd Meas.**		**Dif %**
	** *n* **	**%**	** *n* **	**%**		** *n* **	**%**	** *n* **	**%**		** *n* **	**%**	** *n* **	**%**		** *n* **	**%**	** *n* **	**%**	
None	16	19	3	4.9	−14	32	39	17	28	−11	18	22	39	64	42	7	8.4	1	1.6	−7
Slight	22	27	7	12	−15	25	30	18	30	−1	30	36	16	26	−10	28	34	1	1.6	−32
Some	19	23	29	48	25	16	19	17	28	8.6	13	16	6	9.8	−6	26	31	13	21	−10
High	21	25	19	31	5.8	4	4.8	7	12	6.7	16	19	0	0	−19	19	23	27	44	21
Very high	3	3.6	3	4.9	1.3	2	2.4	1	1.6	−1	3	3.6	0	0	−4	2	2.4	19	31	29
n	81	100	61	100	0	79	100	60	100		80	100	61	100		82	100	61	100	0

**Table 4 ijerph-19-02712-t004:** Comparison between samples of health care professionals through effect size.

**1st Meas. Clinical Hospital of Salamanca**		** *n* **	**Mean**	**SD**	**CV**	**Cohen’s d**	**Hedge’s g**	**Δ_2_ Glass**	**R^2^**
		79	10.91	5.46	50.05				
Other studies	Hospital 12 de Octubre	100	7.83	5.29	67.56	0.57	0.57	0.58	0.08
	Control	70	6.32	4.28	67.72	0.92	0.92	1.07	0.16
2nd Meas. Clinical Hospital of Salamanca		59	7.88	2.89	36.67				
Other studies	Hospital 12 de Octubre					0.01	0.01	0.01	<0.001
	Control					0.42	0.41	0.36	0.001
1st Meas. and 2nd Meas. Clinical Hosp. of Salamanca						0.66	0.66	0.55	0.13

**Table 5 ijerph-19-02712-t005:** Frequency of BAI test results.

BAI Results	Meas. 1		Meas. 2		Dif %
	*n*	%	*n*	%	
Non-anxiogenic (0–9)	33	42.9	33	56.9	14.00
Moderate anxiety (10–18)	18	23.4	15	25.8	2.46
Moderate to severe anxiety (19–29)	20	26.0	9	15.5	−10.48
Severe anxiety (30–63)	6	7.8	1	1.7	−6.08
Total	77	100	58	100	

**Table 6 ijerph-19-02712-t006:** Effect size between BAI factors. First measurement.

Measures	Factors	*n*	Mean	SD	Comparison	Cohen’s d	Hedge’s g	Glass’s Δ_2_
1st Meas.	F1	78	6.22	5.14	F1 and F2	0.33	0.33	0.32
	F2	77	7.98	5.49				
	Total	77	14.32	10.30	1st and 2nd Meas.	0.29	0.29	0.37
2nd Meas.	F1	60	5.08	3.74	F1 and F2	0.38	0.38	0.36
	F2	59	6.57	4.04				
	Total	58	11.58	7.38				

Note. F1: Anxiety Factor 1 (somatic symptoms); F2: Anxiety Factor 2 (subjective and panic symptoms); BAI: Beck Anxiety Inventory.

**Table 7 ijerph-19-02712-t007:** Difference in means of BDI-II scores, before (Dep1) and after (Dep2).

No.	Item	Mean_Dep1_ (SD)	Mean_Dep2_ (SD)	Dif
16	Changes in sleep patterns	1.32 (0.80)	0.82 (0.62)	−0.50
21	Loss of sexual interest	0.90 (0.92)	0.46 (0.74)	−0.44
1	Sadness	0.67 (0.70)	0.30 (0.49)	−0.37
10	Crying	0.59 (0.63)	0.23 (0.50)	−0.36
18	Changes in appetite	0.81 (0.76)	0.57 (0.67)	−0.24
19	Difficulty concentrating	1.00 (0.76)	0.77 (0.62)	−0.23
4	Loss of pleasure	0.83 (0.70)	0.61 (0.59)	−0.22
11	Irritability	0.72 (0.48)	0.51 (0.50)	−0.21
20	Tiredness or fatigue	1.04 (0.76)	0.85 (0.68)	−0.19
3	Failure	0.53 (1.02)	0.43 (0.62)	−0.10
5	Guilt	0.47 (0.64)	0.42 (0.59)	−0.05
8	Self-criticism	0.69 (0.65)	0.64 (0.66)	−0.05
12	Loss of interest	0.74 (0.75)	0.69 (0.74)	−0.05
15	Loss of energy	1.00 (0.65)	0.95 (0.76)	−0.05
14	Devaluation	0.53 (0.85)	0.52 (0.72)	−0.01
2	Pessimism	0.79 (0.67)	0.80 (0.65)	0.01
13	Indecisiveness	0.55 (0.62)	0.57 (0.72)	0.02
17	Agitation	0.79 (0.59)	0.82 (0.65)	0.03
7	Self-dislike	0.36 (0.66)	0.44 (0.70)	0.08
9	Suicidal ideation	0.01 (0.11)	0.10 (0.30)	0.09
6	Punishment	0.22 (0.58)	0.33 (0.79)	0.11

**Table 8 ijerph-19-02712-t008:** Frequency of BDI-II test results.

DBI-II Results	1st Meas.		2nd Meas.		Dif
	*n*	%	*n*	%	
Minimal depression (0–13)	35	46.7	35	58.3	11.63
Mild depression (14–19)	22	29.3	16	26.7	−2.63
Moderate depression (20–28)	13	17.3	5	8.3	−8.97
Severe depression (29–63)	5	6.7	4	6.7	−0.03
Total	75	100	60	100	

**Table 9 ijerph-19-02712-t009:** Frequencies and differences in the satisfaction dimension in both measurements.

Satisfaction	Satisfaction 1st Meas.						Satisfaction 2nd Meas.								
	Personal		Job				Personal			Job					
			Current		Previous					Current			Previous		
	*n*	%	*n*	%	*n*	%	*n*	%	Dif	*n*	%	Dif	*n*	%	Dif
Dissatisfied	1	1.3	10	12.8	0	0	0	0	−1.3	4	6.6	−6.3	1.0	1.6	1.6
Not very satisfied	13	16.7	19	24.4	10	12.8	7	11.5	−5.2	20	32.8	8.4	5.0	8.2	−4.6
Normal	22	28.2	15	19.2	17	21.8	22	36.1	7.9	17	27.9	8.6	18.0	29.5	7.7
Satisfied	31	39.7	26	33.3	22	28.2	26	42.6	2.9	16	26.2	−7.1	30.0	49.2	21.0
Very satisfied	11	14.1	8	10.3	29	37.2	6	9.8	−4.3	4	6.6	−3.7	7.0	11.5	−25.7
Total	78	100	78	100	78	100	61	100		61	100		61	100	

**Table 10 ijerph-19-02712-t010:** Frequency of substance use.

Use	Benzodiazepines	%	Alcohol	%	Tobacco	%
Never	42	68.9	46	75.4	53	86.9
Started	6	9.8	4	6.6	1	1.6
Decreased	3	4.9	3	4.9	3	4.9
Stable	8	13.1	7	11.5	3	4.9
Increased	2	3.3	1	1.6	1	1.6
n	61	100	61	100	61	100

**Table 11 ijerph-19-02712-t011:** Bivariate correlations using Pearson’s correlation coefficient.

	Satisfaction			Insomnia	Depression	Anxiety			Substances		
	Personal	Current Job	Previous Job			Total	F1	F2	Benzo	Alcohol	Tobacco
Personal satisfaction	1										
Current job satisfaction	0.55 **	1									
Previous job satisfaction	0.32 **	0.36 **	1								
Insomnia	−0.26 **	−0.29 **	NS	1							
Depression	−0.44 **	−0.57 **	−0.26 **	0.59 **	1						
Anxiety	−0.43**	−0.47 **	NS	0.58 **	0.69 **	1					
Somatic symptoms of anxiety	−0.38 **	−0.39 **	NS	0.54 **	0.65 **	0.949 **	1				
Subjective anxiety and panic symptoms	−0.47 **	−0.53 **	NS	0.57 **	0.69 **	0.953 **	0.809 **	1			
Benzodiazepines	−0.30 *	NS	NS	0.37 **	0.33 **	0.497 **	0.379 **	0.556 **	1		
Alcohol	NS	−0.34 **	−0.38 **	NS	NS	NS	NS	NS	NS	1	
Tobacco	NS	NS	NS	NS	NS	NS	NS	NS	NS	NS	1

Note. * *p* < 0.05; ** *p* < 0.01. NS: non-significant.

**Table 12 ijerph-19-02712-t012:** Characterization of the segments according to anxiety, depression, and insomnia. Percentages.

**Cluster**	**Anxiety**				
	**Non-anxiogenic**	**Moderate**	**Moderate to severe**	**Severe**	**Total**
1	95.0	5.0	0.0	0.0	100
2	44.4	46.3	9.3	0.0	100
3	0.0	11.8	67.6	20.6	100
Total	48.4	24.2	21.9	5.5	100
**Depression**					
	**Minimal/non-existent**	**Mild**	**Moderate**	**Severe**	**Total**
1	97.5	2.5	0.0	0.0	100
2	48.1	38.9	7.4	5.6	100
3	8.8	35.3	41.2	14.7	100
Total	53.1	26.6	14.1	6.3	100
**Insomnia**					
	**Clinical absence**	**Subthreshold**	**Moderately severe**	**Severe**	**Total**
1	85.0	15.0	0.0	0.0	100
2	22.2	70.4	7.4	0.0	100
3	5.9	38.2	55.9	0.0	100
Total	37.5	44.5	18.0	0.0	100

Note. Percentage belonging to each mental health segment to show the behavior within each cluster and the severity of mental health conditions.

## Data Availability

The data presented in this study are available upon request from the corresponding author. Data have not been made public due to ethical issues.

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
