# Peer review of "Impact on Sleep Quality, Mood, Anxiety, and Personal Satisfaction of Doctors Assigned to COVID-19 Units"

_ijerph, 2022, doi:10.3390/ijerph19052712_

Round 1

Reviewer 1 Report

The study is interesting. But the article needs to be reworked. What is the main objective? What are the secondary objectives? There are missing elements in the methodology. The presentation of the results needs to be revised to be clearer.

Introduction:
There is a mismatch between the aim of the objective and the results presented which are far too broad

Method :
Ethics: usually, authors put the number of the ethics committee's opinion. I am surprised that the committee accepted a study done on google form, which does not respect the European regulation on data protection, especially when it comes to health data.

ISI: the thresholds chosen are not mentioned in the methods section

BAI: I think there is an inversion between ref 16 and 22 to justify the scores chosen for the BAI. The justification is based on an article not referenced on pubmed (J. Sanz, "Recommendations for the use of the Spanish adaptation of the Beck Anxiety Inventory (BAI) in clinical practice," 437 Clínica y Salud,). Do the authors have another reference to justify the thresholds chosen?

BDI: the thresholds chosen are not listed in the methods section

Statistical analyses: what tests were used for descriptive statistics?

Results:
 The population is not described: how many nurses, doctors, what percentage of men, average age,... A table is missing. Characteristics of non-respondents?
The presentation of the results is confusing: there should be a large table with all the scales for the 2 waves to have an overall view. And then detail for each sub-scale

Table 4 should be located between tables 3 and 5 and not at the end

The relevance of making clusters on a small sample size is not discussed

Discussion :
Should be modified
First, synthesis of the results and after  comparison with the literature. Justification of the choices made, particularly in terms of statistical analysis and limitations.
The sample size should be included in the limits

Conclusion:
The conclusion is 40 lines long. This is too long.

Author Response

We have responded to both reviewers, and now resubmit the manuscript with the changes made, hoping that it can be published by your journal. 

Thank you very much

Best regards. 

Reviewer 2 Report

Thank you for the opportunity to review this manuscript titled, “Impact on Sleep Quality, Mood, Anxiety and Personal Satisfaction of Doctors Assigned to COVID-19 Units.” The aim of this study was to describe levels of insomnia, anxiety, depression, and the impact on quality of life of healthcare professionals who were on the frontline of COVID-19 care during the first two waves of the pandemic. Please see detailed comments and suggestions below:

  • The title is very misleading. The title says “doctors” but the manuscript describes a survey of “health care professionals.” Please clarify and use consistent terminology throughout.
  • Using words like “significant” in the abstract is not advisable as it implies a statistical association which is not supported with inferential analysis. Please delete this term, and instead use objective descriptors like “the majority..” or something similar. If inferential analyses were used, please include appropriate supporting values like p-values and/or CIs.
  • The introduction feels disjointed. It would make more sense to frame the introduction in terms of the known stress of the pandemic on healthcare providers (lots of citations available), and not necessarily the health consequences of insomnia. Paragraph 3 is confusing. Poor sleep quality is not the same as insomnia, and from your statements, it doesn’t seem that insomnia is any more prevalent among COVID healthcare workers compared to the general population. Is that the intent?
  • Please clearly state the study design. This appears to be a cross-sectional study, however the survey could have been taken by the same participants twice, but it may not have been the same cohort taking the survey twice? It seems you sent this survey out to 110 people twice, and 1 time 83 people took it, and 1 time 61 people took it? Please clarify.
  • Please describe the hospital setting. Where did the study take place, is there a sense of the volume of patients in these units? Were COVID patients only admitted to the units involved in this study, or were they in other parts of the hospital as well?
    • For Fig 1, please label the axes. I don’t think inclusion of male vs female patients is necessary. Does exitus mean survived or died?
  • Please clarify what is meant in the participants section by “healthcare professionals” is this physicians, nurses, etc?
  • Please clarify the ethical approval for this study. “The ethics committee was informed in advance” is not sufficient.
  • Please clarify why such a short window of time for data collection in April, and such an extended period of time for the second data collection period (Aug to Nov). I am not sure you can compare a 1-week window to a 3-month window?
  • The description of the statistical analysis is not clear. I did not understand how the Health Care Complex control group fit into the overall aims of the study, or how each data collection point was handled within the principal component analysis?
    • What other subject characteristics were included in the analysis? Certainly age, years of experience, shifts worked, etc. are important considerations in interpreting the results.
    • Please justify use of the segment analysis and why you are classifying insomnia as a mental health condition.
    • How was missing data handled, how many subjects had no missing data that you included in the segment analysis?
  • In the results section, it would be helpful to include numeric values when describing data. For example, this sentence, “The quality of sleep improved, reaching similar values to those of other samples of health workers (not exclusively doctors)[27], but not as high as those of the  general population” would be much clearer with numeric values.
    • There is absolutely no description of demographic characteristics of the sample. This is a major limitation in interpreting the study.
  • Overall, there are a lot of tables and figures. For example, table 13 seems unnecessary. It may be worth considering supplemental tables/figures for ease of reading.
  • The overall discussion largely restates the results of the study. I am left unsure of how this paper fits within the large body of evidence related to healthcare worker stress during the pandemic. What is the takeaway from this study?
    • One missing limitation is related to sampling bias. At least from what I gathered from the methods, it is not necessarily the same subjects taking the survey twice. It may be, for example, that the subjects who increased in insomnia/depression/anxiety etc were too stressed to participate in the study the second time, leading to sampling and response bias. Please include this limitation in the discussion section.

Author Response

(The authors gave the same response as above.)

Round 2

Reviewer 1 Report

Authors answer to all my points.

Reviewer 2 Report

The authors have made changes based on reviewers comments.